# Multi-Feature Fusion Method Based on Adaptive Dilation Convolution for Small-Object Detection

**DOI:** 10.3390/s25103182

**Published:** 2025-05-18

**Authors:** Lin Cao, Jin Wu, Zongmin Zhao, Chong Fu, Dongfeng Wang

**Affiliations:** 1School of Information and Communication Engineering, Beijing Information Science and Technology University, Beijing 100101, China; charlin@bistu.edu.cn (L.C.); wujin99@bistu.edu.cn (J.W.); 2Key Laboratory of the Ministry of Education for Optoelectronic Measurement Technology and Instrument, Beijing Information Science and Technology University, Beijing 100101, China; 3School of Computer Science and Engineering, Northeastern University, Shenyang 110169, China; fuchong@mail.neu.edu.cn; 4Beijing TransMicrowave Technology Company, Beijing 100080, China; wdf@tsmtc.com

**Keywords:** small-object detection, attention mechanism, feature fusion, adaptive dilated convolution, radar–camera system

## Abstract

This paper addresses the challenge of small-object detection in traffic surveillance by proposing a hybrid network architecture that combines attention mechanisms with convolutional layers. The network introduces an innovative attention mechanism into the YOLOv8 backbone, which effectively enhances the detection accuracy and robustness of small objects through fine-grained and coarse-grained attention routing on feature maps. During the feature fusion stage, we employ adaptive dilated convolution, which dynamically adjusts the dilation rate spatially based on frequency components. This adaptive convolution kernel helps preserve the details of small objects while strengthening their feature representation. It also expands the receptive field, which is beneficial for capturing contextual information and the overall features of small objects. Our method demonstrates an improvement in Average Precision (AP) by 1% on the UA-DETRAC-test dataset and 3% on the VisDrone-test dataset when compared to state-of-the-art methods. The experiments indicate that the new architecture achieves significant performance improvements across various evaluation metrics. To fully leverage the potential of our approach, we conducted extended research on radar–camera systems.

## 1. Introduction

In the evolving landscape of intelligent transportation systems, the ability to accurately and proactively detect potential threats in traffic scenes is a critical factor in ensuring road safety. This is especially important in high-density urban environments, complex road structures, and scenarios that require all-weather operations. Among these challenges, the detection of small, distant objects in images plays a particularly crucial role [1]. Such objects typically refer to traffic participants or items that are small in size, have low texture, and are prone to occlusion, such as distant electric scooters, pedestrians, or non-motorized vehicles [2]. The early detection of these targets is vital for achieving key tasks such as forward collision warning, path planning, and emergency avoidance [3]. Accordingly, this study focuses on the image-based detection of small, distant traffic-related objects, aiming to enhance visual systems’ perceptions of these high-risk, subtle targets and thereby better support a wide range of traffic safety tasks.

Improving the detection of small objects can yield substantial safety benefits in real-world applications. For example, surveillance cameras mounted on urban overpasses or intersections can issue early warnings by detecting distant approaching electric bikes or pedestrians in a timely manner, thereby assisting traffic control systems in rapid decision-making [4]. In addition, detecting small, distant objects at intersections helps accurately assess traffic flow conditions, enabling the early identification of queue build-up and abnormal congestion. This information supports traffic signal control and congestion-mitigation strategies. In autonomous driving, accurately recognizing distant, partially occluded vehicles or low-visibility traffic participants can significantly reduce collision risks by enabling earlier maneuver adjustments. Furthermore, under nighttime or adverse weather conditions, such as rain, snow, or fog, the reliable detection of distant headlights and vehicle contours can enhance the stability of multi-sensor fusion systems [5,6], improving the overall robustness of environmental perception. These scenarios demonstrate that the core concern is not merely the object category itself, but rather its safety-critical implications within the context of traffic.

Despite the widespread adoption of existing detection models such as the YOLO series [7], R-CNN [8], and DETR [9], these approaches still face significant challenges in detecting small and distant objects in the aforementioned complex scenarios. Due to their small pixel footprint, sparse features, and low contrast against the background, small objects are often prone to information loss during downsampling, resulting in missed or false detections. Additionally, the presence of cluttered backgrounds, occlusions, and scale variations further hampers model performance in terms of stability and generalization. Conventional models also exhibit limited capacity for contextual reasoning, which makes it difficult to model semantic relationships between objects and their surrounding environment. This capability is essential in real-world traffic scenarios.

To address these problems, researchers have proposed various solutions. Li et al. [10] conducted an in-depth study on how receptive fields affect multi-scale object detection and proposed an adaptive multi-scale detection framework. The framework employs parameter sharing and applies different dilated convolutions across three branches to mitigate the impact of scale variation. However, this approach still fails to adequately address the significant size variations of targets within the same feature map. Compared with this approach, the proposed method can adaptively adjust the dilation rates of dilated convolutions, effectively reducing the scale disparity of targets within the same feature map. Bai et al. [11] recognized that small-scale objects inherently lack sufficient distinctive features, making them particularly challenging to distinguish from background clutter, compounded by their typically low-resolution and blurry appearance. To address this, they developed MTGAN, an end-to-end multi-task generative adversarial network for small-object detection. This framework employs a generator network to produce super-resolved images, while incorporating a multi-task discriminator network that simultaneously performs three functions: authenticating genuine high-resolution images, predicting object categories, and refining bounding boxes. However, this approach presents several limitations: It requires the pre-generation of images that demand substantial computational resources, exhibits insufficient modeling capability for fine details of small objects, and tends to introduce texture distortions. In contrast, the proposed method eliminates the generation step, thereby significantly conserving computational resources while fully leveraging the original image data. Ref. [12] proposed a novel vision transformer architecture incorporating binary-level routing attention mechanisms, which significantly enhances visual task performance through hierarchical feature processing. However, this approach exhibits a critical limitation: Its attention mechanism exclusively focuses on the highest-scoring regions in the adjacency matrix. Such excessive selectivity may lead to the omission of semantically relevant but lower-scoring information and compromised model generalization capability. In contrast, our method systematically extends attention coverage to secondary relevant regions, effectively addressing these limitations and consequently achieving superior generalization performance. Kisantal et al. [13] identify the scarcity of small objects in datasets as a primary factor contributing to performance degradation. Their proposed augmentation strategy combines the oversampling of images containing small objects with copy–paste operations for small targets. While this approach effectively increases small-object samples, it introduces inherent limitations: The naive copy–paste operation disrupts spatial coherence in images, and direct resizing generates unrealistic textures while distorting original aspect ratios. In contrast, the proposed method preserves the spatial coherence and natural physical layout of the original images without artificial disruption, making it more consistent with human cognitive patterns.

This paper addresses the challenge of detecting small, distant, low-feature, and high-risk traffic objects by proposing a novel method that combines feature enhancement and perceptual fusion. The approach introduces an adaptive receptive field mechanism and a spatial frequency joint modeling strategy to improve the representation of weak object features. During the feature extraction phase, a lightweight attention module is employed to model long-range contextual dependencies, enhancing the semantic association between small objects and their environments. During feature fusion, frequency-domain decomposition is utilized to supplement structural information and strengthen the modeling of blurry contours. Additionally, this work utilize the proposed visual detection method to assess the sensing capability of millimeter-wave radars in detecting distant targets. By comparing visual detection results with radar outputs, we analyze the accuracy and stability of radar in recognizing small distant objects in complex traffic scenes, providing valuable insights for the optimization of future multi-sensor fusion strategies. The contributions of this paper are as follows:Based on the YOLOv8 object detection architecture, a new feature fusion architecture is proposed, which consists of three multi-scale feature fusion modules, a triple-feature encoding module, and a downsampling process, that can fuse the features of different scales from the backbone network. The proposed architecture is used to verify the accuracy of radar detection, laying the foundation for the post-fusion technology of radar and cameras.A new triple-feature coding module is proposed, which performs different processing on the features of three different scales. Adaptive dilated convolution is applied to features of different frequency bands (corresponding to object scales) with varying dilation rates, reducing intra-class size variance in feature maps and enabling dynamic receptive field adaptation.A new self-attention module is proposed and added to the original CSPDarknet backbone network, which captures global dependencies through a token-to-token self-attention mechanism to bring about better context analysis, resulting in the learning of more distinguishable feature representations.

The remainder of this paper is organized as follows: Section 2 mainly introduces related works in small-object detection. In Section 3, we provided a detailed description of the proposed method. In Section 4, we introduce our experiments and related analysis. This work is concluded in Section 5.

## 2. Related Work

In this section, the most relevant research is presented, encompassing small-object detection, attention mechanisms, and feature fusion networks.

### 2.1. Small-Object Detection

Small-object detection has attracted extensive research. Cheng et al. [14] have conducted a comprehensive survey on small-object detection technologies, which are roughly divided into the following: sampling-oriented methods, scale-aware methods, attention-based methods, feature mimicry methods, and density cropping methods. Due to the dense occurrence of small objects in high-resolution images, density cropping is a popular strategy in small-object detection. Meethal et al. [15] propose a cascaded zoom-in detector specifically for small-object detection in high-resolution aerial images. It utilizes density cropping technology to extract and process crowded small-object regions in images, thereby improving detection accuracy. During the training phase, the detector itself is repurposed for density-guided training and inference. In the inference phase, it first detects base-class objects and performs density cropping, and then, it employs the cropped regions for more refined reasoning in the second stage. However, cropping low-density areas may lead to the neglect of important contextual information, and since only high-density areas are focused on, other areas that may contain small objects but have lower densities are easily overlooked, resulting in false positives. Due to its two-stage reasoning, the detection speed cannot not meet the requirements of real-time applications. CenterNet [16] is an efficient object detection algorithm that achieves detection by transforming the object detection problem into a keypoint estimation problem. It does not rely on predefined anchor boxes but directly predicts the center point, width, and height of the object from the feature map. Even if part of the object is occluded, as long as the center point is visible, CenterNet can still accurately detect the object, reducing the risk of false positives due to inaccurate anchor box matching. However, the center point of small objects is often difficult to locate precisely in the image, and in situations with a large number of small objects or complex backgrounds, CenterNet may be more affected by background noise, increasing the risk of false positives. In contrast, the proposed method uses one-stage inference and has higher real-time detection. At the same time, the proposed feature fusion method has higher accuracy for small-object detection.

### 2.2. Attention Mechanism

Attention mechanisms have become a key component in various fields of artificial intelligence. In the field of object detection, attention mechanisms have been employed to enhance the detection of small objects. For example, VSTAM [17] utilizes the Video Sparse Transformer to process video frames, capturing spatiotemporal information through sparse convolutions. An attention-guided memory mechanism is then introduced to enhance the feature extraction and memory of key objects in the video. Finally, pixel-level enhancement is applied to improve the detection accuracy of object candidate regions. Despite the use of sparse processing, the implementation may still suffer from high computational complexity, especially when dealing with high-resolution videos. Huang et al. [18] addressed three critical limitations of the DETR framework—slow convergence speed, instability in its one-to-one matching mechanism, and the challenge of accurately localizing and classifying small targets in dense scenes—by introducing ground-truth annotations as supervisory signals during the initial training phase. This approach guides the model to learn more appropriate matching relationships while integrating a one-to-many matching strategy, thereby resolving the inherent limitations of the original DETR framework. Perreault et al. [19] first generates foreground and background segmentation labels in a semi-supervised manner, applies an attention mechanism to the segmentation labels, and then weights the feature maps used to generate bounding boxes to reduce the signal from non-relevant areas. This allows the model to dynamically focus on the most important regions of the image, thus better capturing the target and its contextual information. However, the performance of this method is affected when dealing with densely distributed small objects. TPH-YOLO [20] adds a prediction head based on self-attention mechanisms to the traditional YOLOv5, allowing the model to better capture global contextual information. Feature maps from different levels are fed into the Convolutional Block Attention Model (CBAM) to help the model identify attention areas in dense scenes. Due to the introduction of the transformer prediction head and additional prediction heads, the computational and memory overhead of TPH-YOLO significantly increases. The performance improvement of TPH-YOLO is primarily focused on drone-captured scenarios, and its generalization ability in other scenarios has not been fully validated.

### 2.3. Feature Fusion

One of the main difficulties in object detection is effectively representing and processing scale features. Early detectors typically made predictions directly based on the pyramid feature hierarchy extracted from the backbone network. PANet [21] introduces a bottom–up path to enhance the precise localization information in lower-level features, which helps improve the information flow of the entire feature pyramid and shortens the information path from the bottom to the top features. This approach allows each proposal to obtain information from all feature levels, thus avoiding reliance solely on a manually specified feature level. Although PANet significantly improves performance on instance segmentation tasks, it does not perform well when dealing with densely distributed small objects. In FFAVOD [22], the feature fusion module allows the network to share feature maps between adjacent frames, reducing the need for the separate processing of each frame and enhancing temporal continuity. Chen et al. [23] proposed a frequency-adaptive dilated convolution to deal with artifacts in segmentation tasks. Instead of using a fixed global expansion rate, it dynamically adjusts the expansion rate spatially according to the local frequency components.

### 2.4. Radar and Camera System

Radar object detection plays a crucial role in radar–camera systems. Yang et al. [24] proposed a CFAR algorithm based on the Monte Carlo method, aiming to enhance the performance of object detection in complex road environments. This method utilizes Monte Carlo random sampling to estimate the statistical characteristics of background noise and clutter by simulating a large number of random samples. Xu et al. [25] proposed a radar-based algorithm for the automatic detection of Foreign Object Debris (FOD) on airport runways, aiming to improve the detection of small objects at long distances. Qin et al. [26] have further developed an improved Hausdorff distance matching algorithm for fusing video information from cameras with radar data. This approach is designed to align and analyze the correlation of multi-object data in spatial dimensions and to match highly similar data to determine the Regions of Interest (RoIs) for vehicle detection.

## 3. Methodology

The proposed network architecture is a hybrid model based on YOLOv8, which uses a convolution layer (Conv) and self-attention. The architecture is shown in Figure 1, which consists of three main parts. The first part is the backbone, which integrates the proposed Skipped Binary-Routing Attention (SBRA) into the original CSPDarknet to extract more differentiated features. The second part is the bottleneck layer, which adopts a new Multi-Feature Fusion Network instead of a PANet to fuse high-level and low-level features. Moreover, adaptive dilated convolution is introduced to balance the receptive field and effective bandwidth. The last part is the detection header part, which outputs the detection results through object classification and regression. These three components will be elaborated on in detail in the following subsections.

### 3.1. Backbone

Images captured by surveillance cameras are characterized by large-scale and complex scenes. It is difficult to extract the overall information of small objects because the feature information of CNNs is seriously lost after multi-layer convolution and pooling operations. The research in article [27] shows that the attention mechanism is more robust to severe occlusion, perturbation, and domain shifts than CNNs. In this paper, a new attention mechanism is integrated into the backbone network of YOLOv8 to improve semantic discriminability and gather scene information from extensive regions. This integration enables the backbone to simultaneously attend to local features and long-distance contextual information.

Our network is a hybrid model based on YOLOv8, for which its backbone consists of part of the YOLOv8 backbone and the improved BRA. Among them, the YOLOv8 backbone contains 5 stages, called YOLOv8-p5. These 5 stages are usually called [p1, p2, p3, p4, p5], which have the downsampling multiplicity of [2, 4, 8, 16, 32] with respect to the input image, respectively. Except for the stages of P1 and P2, the feature maps pass through the C2f module before entering the [P3, P4, P5] layers. The number of C2f modules before stages [P3, P4, P5] is [3, 6, 9]. The C2f module is shown in Figure 2a, which consists of three Convs and one Bottleneck (as shown in Figure 2b). CNNs are responsible for extracting low-level local features, such as edges and textures, from the image. These features are then fed into the SBRA module, which captures global dependencies through a token-to-token self-attentive mechanism, thus effectively modeling long-range dependencies and complex contextual information in the image. By combining the strengths of these two components, the model is able to enhance the understanding of global information and improve the task performance while maintaining local feature awareness. Applying attention mechanism on lower resolution feature maps can reduce computational costs and memory, so in this article, SBRA is between P5 and SPPF because the feature map resolution of this layer is lower.

The input of SBRA can be obtained by the following formula:(1)$$\begin{array}{c} Y=G_{\sigma }\times X \end{array}$$
where $X\in R^{H\times W\times c}$ represents the input image, which is a three-dimensional tensor. *H* and *W* represent the height and width of the image, respectively, and c denotes the number of channels in the image. $G_{\sigma }$ represents a series of convolution layers using 2D Gaussian filters: that is, the convolution layers of P1 to P5. The $G_{\sigma }$ is shown as follows:(2)$$\begin{array}{c} G_{\sigma }=\frac{1}{2\pi \sigma^{2}}e^{-\left(W^{2}+H^{2}\right)/2\sigma^{2}} \end{array}$$
where σ is the scaling parameter of the standard deviation of the 2D Gaussian filter used for convolution.

The feature map Y can be obtained through the five-stage Conv and C2f modules, for which their height and width are *h* and *w*. Divide Y into S×S non-overlapping areas and obtain a total of $\frac{h\times w}{S^{2}}$ feature vectors, as shown in Figure 3. The query $Q_{r}$, key $K_{r}$, and value $V_{r}$ tensors can be derived by linear projection:(3)$$\begin{array}{c} \left\{\begin{array}{c} Q_{r}=Y_{r}W_{q}\\ K_{r}=Y_{r}W_{k}\\ V_{r}=Y_{r}W_{v} \end{array}\right. \end{array}$$
where $Q_{r},K_{r},V_{r}\in R^{S^{2}\times \frac{hw}{S^{2}}\times c}$ and $W_{q},W_{k},W_{v}\in R^{c\times c}$ are projection weights for the query, key, and value, respectively.

The contextual relationship between patches is then found by constructing a directed graph. Specifically, it is first performed by applying each region-level query operation to $Q_{r},K_{r}$. Then, we derive the adjacency matrix $A_{r}\in R^{S^{2}\times S^{2}}$ of the region-to-region affinities by matrix multiplication between $Q_{r}$ and transposed $K_{r}$.(4)$$\begin{array}{c} A_{r}=Q_{r}{\left(K_{r}\right)}^{T} \end{array}$$

The $A_{r}$ adjacency matrix measures the semantic relevance of the two regions.

It should be noted that only the first *k* most relevant regions are focused on in the BRA method. It has been observed that focusing too much on the most relevant region may cause the following problems. It may cause the model to overlook information that is less relevant but still important, which can lead to a decline in performance when dealing with complex tasks. Secondly, the model may become overly reliant on the most relevant features in the training data, leading to a decrease in the model’s generalization ability. The model is too sensitive to noise or small changes in the input data because it focuses on a small region. Insufficient dependencies over long distances may not be captured effectively. In multi-step tasks, if a mistake is made at an early stage by focusing only on the most relevant region, this mistake will accumulate in the subsequent process and lead to a large deviation in the final result.

To solve this problem, we generate the path index matrix $I_{r}$ by retaining the first *k* connections of each region in $A_{r}$ and the *k* connections spaced between the remaining connections. The token-to-token attention can then be realized by collecting the key–value tensor through the route index matrix. Specifically, each row of $I_{r}$ represents one operator, totaling 2k operators. For each query token in region *i*, it will process all key–value pairs residing in the 2k paths indexed by $\left\{I_{i,1}^{r},I_{i,2}^{r}\ldots I_{i,2k}^{r}\right\}$. The method for creating the path index matrix and collecting the key–value tensor is shown below:(5)$$\begin{array}{c} \left\{\begin{array}{c} I_{r}=topkIndex\left(A_{r}\right)+skipkIndex\left(A_{r}\right)\\ K_{g}=gather\left(K,I_{r}\right)\\ V_{g}=gather\left(V,I_{r}\right) \end{array}\right. \end{array}$$
where $K_{g},V_{g}\in R^{S^{2}\times \frac{kHW}{S^{2}}\times c}$ is the collected key–value tensor. The topkIndex operation represents the selection of indices for the most relevant patches in the adjacency matrix $A_{r}$, while the skipkIndex operation denotes the selection of indices for paths at equal intervals within $A_{r}$. The gather function searches for image paths based on $I_{r}$. Finally, through the Attention operation, output O can be obtained:(6)$$\begin{array}{c} O=Attention(Q,K_{g},V_{g}) \end{array}$$

### 3.2. Neck

In an increasingly deep backbone network, the size of the feature maps continuously decreases, and the number of object features becomes progressively fewer [28]. Additionally, within the same feature map, there is a significant discrepancy in the quantity of object features across different scales. This results in a significantly smaller loss for small objects compared to large objects, which is detrimental to the detection of small objects.

In order to deal with the problems mentioned above, we propose a new feature fusion network. It can better combine the high-dimensional information of deep feature maps with the detailed information of shallow feature maps and reduce the difference in the number of features between objects of different scales in the same feature map. Its architecture is shown as Neck in Figure 1, which mainly consists of three cascaded Scale Fusion Feature (SFF) modules, a TFAS module, and a downsampling network.

There are a total of three SFF modules in Neck, which are connected in series. Each SFF module is composed of four components: a convolutional layer, upsampling, the TFA module, and the C2f module. Each SFF module includes two inputs and two outputs. The SFF(a) module receives one input from the SPPF in the backbone and another input from the layer of P4. One output of SFF(a) is fed into SFF(b), while the other output is concatenated with the 32nd layer. SFF(b) not only receives the output features from SFF(a) but also takes input features from the fourth layer of the backbone, which is P3. The first output of SFF(b) is passed to SFF(c), and the other output is concatenated with the 28th layer. Similarly, SFF(c) receives input features from SFF(b), as well as from the second layer of the backbone P2. The first output of SFF(c) is concatenated with the output of TFAS and then enters the top–down network, while the second output is concatenated with the 25th layer. The initial operation of the SFF module is a convolution layer that takes the output features from the SPPF layer of the backbone as input. One branch of this convolution is directly used as the first feature input for the TFA layer, and another branch is upsampled to serve as the second input for TFA. The output of the TFA module is processed by C2f before being passed to the next layer of the network.

The TFA module, as depicted in Figure 4, is a three-feature fusion module equipped with adaptive dilated convolution (AdvDilaConv). It takes as input features of three different scales: large, medium, and small. The small- and medium-sized features are derived from the Conv and upsampling layer within the SFF, respectively. The large-sized feature maps in SFF(a), SFF(b), and SFF(c) originate from the backbone network at layers [p4, p3, p2], respectively. Assuming that the number of channels and the size of the medium-scale feature map are 1c and 1s, respectively, the number of channels and the size of the large-scale feature map would be 0.5C and 2S, while for the small-scale feature map, they would be 2C and 0.5S. Before feature encoding, the number of channels can be adjusted to align with the primary scale feature through the Conv, batch normalization, and SiLU (ConvBNSiLU) operations. After the large-scale feature map undergoes processing by the convolutional module, its number of channels is adjusted to 1C, followed by downsampling using adaptive dilation convolution. It assigns different dilation coefficients to features of varying frequencies during the downsampling process, thereby preserving more details from the large-scale feature map. For the small-scale feature map, the convolutional module is used to adjust the channel count, with nearest neighbor interpolation employed for upsampling. This helps maintain the richness of the local features in low-resolution images, preventing the loss of small-object feature information. In TFA, the medium-scale feature map does not undergo the steps within the yellow dashed box but proceeds directly to the next stage. The steps within the yellow dashed box are specifically designed for small objects in the triple-feature fusion module with adaptive dilation convolution and SBRA (TFAS) and will be elaborated upon later. The three feature maps of different sizes are finally concatenated along the channel dimension, as illustrated below:(7)$$\begin{array}{c} Z=Concat(F_{l},F_{m},F_{s}) \end{array}$$
where Z represents the feature map output by the TFA. Large-, medium-, and small-size feature graphs are represented by $F_{l},F_{m}$, and $F_{s}$, and Z is formed by them in series. The output Z has the same resolution as $F_{m}$ and three times the number of channels of $F_{m}$.

The TFAS module is specifically designed for enhancing the detection of small objects, as illustrated in Figure 4. Its inputs stem from the 2nd, 4th, and 6th layers of the backbone network. In contrast to the TFA module, TFAS incorporates SBRA attention into the medium-sized feature maps. Small-object features are typically more clearer in shallow feature maps, whereas features at shallow layers usually lack semantic information. Applying SBRA attention to the intermediate feature maps adds long-range feature correlations, thereby providing richer semantic information for the detection of small objects.

To further aggregate high-level and low-level features for improved object detection performance, after the SFF module, the feature maps are downsampled via Conv and form lateral connections with the outputs of the SFF module. This approach integrates high-level semantic information and reduces the number of layers from the backbone to the head. We adhere to the definition of PANet, where layers producing feature maps of the same spatial size are considered to be at the same network level. As illustrated in Figure 1, each feature map used for detection acquires a richer semantic feature map produced by the SFF module through lateral connections, as well as higher-resolution feature maps from the top–down network. Every feature map is firstly passed through a 3×3 convolutional layer with a stride of 2 to reduce the spatial size, followed by a lateral connection that merges the output of the SFF block with the downsampled map. The fused feature map is then further processed by the C2f module before being fed into the detection head. In particular, the topmost feature map is also connected to the output from the TFAS module.

In addition, the use of large kernel depth convolutions is an effective way to expand the receptive field and enhance modeling capabilities. However, simply utilizing them at all stages may lead to the contamination of shallow features for detecting small objects while also introducing significant I/O overhead and latency at high resolutions. We therefore introduce adaptive dilation convolution in TFAS. Adaptive dilatation convolution can improve the problem of scale variations in image detection to a certain extent. It can assign different dilation rates to different frequency bands in the image such that the model has differently sized receptive fields for different frequency bands.

The adaptive dilation is obtained based on the frequencies, and it can be divided into three parts: frequency selection, adaptive convolution kernel, and adaptive dilation. Firstly, the root divides the whole feature map into four frequency bands by the frequency mask:(8)$$\begin{array}{c} X_{b}=\left\{\begin{array}{cc} F^{-1}X_{F}, & \mathrm{if}\;\phi_{b}⩽max\left(|u|,|v|\right)<\phi_{b+1}\\ 0, & \mathrm{others} \end{array}\right. \end{array}$$
where $F^{-1}$ represents the inverse Fourier transform. $\phi_{b}$ and $\phi_{b+1}$ are two of the B + 1 predefined frequency thresholds $\left\{0,\phi_{1},\phi_{2},\ldots \phi_{B-1},1/2\right\}$; they are used to extract the corresponding frequency.

By decomposing the convolution kernel into low-frequency and high-frequency components and then introducing dynamic weighting to adjust the frequency response, it becomes possible to dynamically adjust the high-frequency and low-frequency components (i.e., adaptive convolution kernels). This approach allows for the allocation of different dilation rates to different frequency bands. This step can be formalized as follows:(9)$$\begin{array}{c} \left\{\begin{array}{c} W^{\prime }=\gamma_{l}\bar{W}+\gamma_{h}\hat{W}\\ \bar{W}=\frac{1}{K\times K}\sum_{i=1}^{K\times K}W_{i} \end{array}\right. \end{array}$$
where $\bar{W}$ represents the kernel-by-kernel average of W. Its function is a low-pass K×K mean filter, followed by a 1×1 convolution. A higher mean value tends to attenuate high-frequency components, which means that as the mean of $\bar{W}$ increases, the differences within it become smaller, resulting in a reduced ability to resolve high frequencies. $\gamma_{l}$, $\gamma_{h}$ are dynamic weights for each channel, and *l* and *h* represent the low frequency and high frequency, respectively. They can be predicted through a simple and lightweight global pooling and convolutional layer. The network dynamically adjusts the ratio of $\gamma_{l}$ and $\gamma_{h}$ based on the context in the input, enabling the network to focus on specific frequency bands and adapt to the complexity of features in visual models. This dynamic frequency adaptation method enhances the network’s capability to simultaneously capture low-frequency contexts and high-frequency local details. In turn, this increases the effective bandwidth and improves the performance of tasks that require different feature extractions at various frequencies.

The adaptive dilation rate can be realized by frequency selection and an adaptive convolution kernel, which assigns different dilation rates to each pixel. For a region centered at *p* with window size *K*, the pixel value can be expressed as(10)$$\begin{array}{c} Y\left(p\right)=\sum_{i=1}^{K\times K}W_{i}X\left(p+\Delta p_{i}\times \hat{D}\left(p\right)\right) \end{array}$$
where X represents the input feature map, and $W_{i}$ denotes the weight of the convolution kernel at position *i*. $\Delta p_{i}$ refers to the ith position of the predefined grid sampling. $\hat{D}\left(p\right)$ represents the dilation rate as a function of the sampling offset. This dilation rate $\hat{D}\left(p\right)$ can be predicted through a convolutional layer with parameter θ, where the relationship between θ and $\hat{D}\left(p\right)$ is given by Equation (Equation 13), and its derivation is outlined below. Specifically, an ReLU layer is introduced after this convolutional layer to ensure the non-negativity of the dilation rate. This design allows for the maximization of the receptive field while minimizing the loss of frequency information at each pixel.

For a local feature centered on *p* with a window size of *s*, it is called $X^{\left(p,s\right)}$. Its receptive field can be expressed as follows:(11)$$\begin{array}{c} RF\left(p\right)=\left(K-1\right)\times \hat{D}\left(p\right)+1 \end{array}$$(12)$$\begin{array}{c} \mathrm{HP}\left(p\right)=\sum_{H_{\hat{D}\left(p\right)}^{+}}{\left|X_{F}^{\left(p,s\right)}\left(u,v\right)\right|}^{2} \end{array}$$

The receptive field is positively correlated with $\hat{D}\left(p\right)$. Since the frequencies in a set $H_{\hat{D}\left(p\right)}^{+}$ cannot be accurately captured, the lost frequency information can be measured by calculating the high-frequency power HP. As a result, the result of θ can be written as follows:(13)$$\begin{array}{c} \theta =\max_{\theta }\left(\sum \mathrm{RF}(p)-\sum \mathrm{HP}(p)\right) \end{array}$$

Since HP is non-differentiable, it can be achieved indirectly by optimizing D(p). Thus, $\theta^{\prime }$ can be expressed as follows:(14)$$\begin{array}{c} \theta^{\prime }=\max_{\theta^{\prime }}\left(\sum_{p\in {\mathrm{HP}}^{-}}\hat{D}\left(p\right)-\sum_{p\in {\mathrm{HP}}^{+}}\hat{D}\left(p\right)\right) \end{array}$$

${\mathrm{HP}}^{-}$ and ${\mathrm{HP}}^{+}$ denote the pixels with the lowest and highest high-frequency power, respectively. It is possible to select a larger dilation rate at a lower HP and a smaller dilation rate at a higher HP to minimize information loss.

### 3.3. Head

Our architecture contains a total of four detection heads, called head1, head2, head3, and head4; their step size with respect to the input image is [4, 8, 16, 32]. They are constructed in a decoupled way, and each detection head has two branches, which are used for regression and classification. Both their classification and regression branches are composed of three convolutional layers, which are used to predict the categories and regress the detection boxes, respectively. The dimension of the output in the classification branch is the number of categories in the dataset. The output in the regression branch equals four times the number of channels associated with the predefined anchors. These four values represent the coordinates of the center point and the width and height of the target box.

To address the issue of the overdomination of easily categorized samples and category imbalance, focal loss is employed. The formula is as follows:(15)$$\begin{array}{c} L_{cls}=-\alpha_{t}{\left(1-p_{t}\right)}^{\gamma }\log\left(p_{t}\right) \end{array}$$
where γ is the modulation factor, and when set to 2, it can control the imbalance of the number of simple/difficult samples. It is used to reduce the loss function of easy-to-classify samples and make the model pay more attention to difficult-to-classify samples. Here, $p_{t}$ is the model’s predicted probability for each category. For positive probabilities, $p_{t}$ is the model’s predicted probability that the target belongs to the positive category. For the negative category, $p_{t}$ is the model’s predicted probability that the target belongs to the negative category. When $p_{t}$ is close to 1, the model’s prediction for positive samples is very accurate; ${\left(1-p_{t}\right)}^{\gamma }$ is close to 0, and the loss will decrease. When $p_{t}$ is close to 0, the model’s prediction for positive samples is very inaccurate; ${\left(1-p_{t}\right)}^{\gamma }$ is close to 1, and the loss will not decrease significantly. $p_{t}$ is defined as follows:(16)$$\begin{array}{c} p_{t}=\left\{\begin{array}{cc} \hat{y}, & \mathrm{if}\;y=1\\ 1-\hat{y}, & \mathrm{if}\;y=0 \end{array}\right. \end{array}$$

$\alpha_{t}$ is the weight adjustment coefficient of positive and negative samples, which is used to balance positive and negative samples. It can suppress the balance of the number of positive and negative samples. $\alpha_{t}$ is defined as follows:(17)$$\begin{array}{c} \alpha_{t}=\left\{\begin{array}{cc} \alpha , & \mathrm{if}\;y=1\\ 1-\alpha , & \mathrm{if}\;y=0 \end{array}\right. \end{array}$$

To enhance the detection accuracy for small objects, we also adopt the state-of-the-art NWDLoss to constrain the position of the target boxes. For the set of predicted boxes $P=\{p_{1},p_{2},\ldots ,p_{n}\}$ and the set of ground-truth boxes $G=\{g_{1},g_{2},\ldots ,g_{n}\}$, where *n* is the number of boxes, the L1 distance between each predicted box $p_{i}$ and its corresponding ground-truth box $g_{i}$ is calculated as follows:(18)$$\begin{array}{cc} L_{1}\left(p_{i},g_{i}\right) & =\;|x_{i1}-x_{i1}^{*}\left|\;+\;\right|y_{i1}-y_{i1}^{*}|\\ & \;+|x_{i2}-x_{i2}^{*}\left|\;+\;\right|y_{i2}-y_{i2}^{*}| \end{array}$$

Then, the *L*_1_ distance is normalized according to the width and height of the real box to eliminate the influence of boxes of different sizes. Finally, all normalized *L*_1_ distances are averaged:(19)$$\begin{array}{c} L_{obj}=\frac{1}{N}\sum_{i=1}^{N}\frac{L_{1}\left(p_{i},g_{i}\right)}{x_{i2}^{*}-x_{i1}^{*}+y_{i2}^{*}-y_{i1}^{*}} \end{array}$$

The final total loss can be expressed as follows:(20)$$\begin{array}{c} L=\alpha_{1}L_{obj}+\alpha_{2}L_{cls} \end{array}$$
where $\alpha_{1}$ and $\alpha_{2}$ represent the proportions of these two losses, and here, they are set to 0.7 and 0.3.

## 4. Experiments

In this section, we first introduce the two object detection datasets used in the experiment, along with the related evaluation methods. Subsequently, we elaborate on the implementation details of the experiment. Finally, we assess the effectiveness of the proposed hybrid model on the object detection task through experimentation. Specifically, we evaluate our method on two datasets and compare it with other methods. Then, we conduct ablation studies to validate the effectiveness of the proposed modules. In conclusion, we present some visualized results.

### 4.1. Datasets

We experimentally evaluate the effectiveness of our proposed method on two object detection datasets. We use a traffic object detection dataset UA-DETRAC [29] and a popular challenging aerial image detection benchmark dataset VisDrone [30]. The two datasets are captured using very different devices, with UA-DETRAC using a stationary camera to capture each scene and VisDrone using a drone. Although there are not many small objects on UA-DETRAC, it has more complex traffic scenarios, which can verify the generalization performance of our model. We show the detection effects of different methods on these two datasets. In addition, we conduct an ablation study to verify the effectiveness of our proposed fusion method.

Visdrone: This dataset contains a total of 10,209 drone images (6471 images in the training set, 548 images in the validation subset, 1580 images in the test challenge subset, and 1610 images in the test development subset), with a maximum resolution of 2000 × 1500. These objects contain a total of 10 object categories, including the following: pedestrians, people, cars, vans, buses, trucks, motorcycles, bicycles, visor tricycles, and tricycles. The dataset encompasses diverse illumination conditions, including sunny, cloudy, rainy, and foggy weather, along with typical traffic scenarios such as urban streets, highways, intersections, and parking lots. It specifically captures challenging real-world complexities like high-density traffic flows, mixed pedestrian–vehicle interactions, and transient occlusions. Notably, the dataset exhibits class imbalance characteristics, making it particularly valuable as a benchmark for small-objec-detection research.

UA-DETRAC: This dataset is a multi-object detection dataset designed for urban road scenarios, comprising 100 video sequences (over 140,000 frames) with a total of 1.21 million annotated vehicle bounding boxes. The data cover various typical traffic scenarios, including urban roads, intersections, etc., and encompass diverse weather and lighting conditions, such as sunny, rainy, cloudy, and nighttime environments.The primary challenges lie in complex situations like vehicle occlusions and dense traffic flow. The vehicle targets are categorized into four types: car, bus, van, and others.

### 4.2. Evaluation Protocols and Implementation Details

Evaluation Protocols: In the field of object detection, mAP (mean Average Precision) is a core metric for evaluating the performance of detectors. It reflects the overall capability of a model by averaging the precision over multiple IoU (Intersection over Union) thresholds ranging from 0.5 to 0.95 in increments of 0.05. mAP integrates both the detection precision and recall across different object categories, where a higher mAP indicates a model’s ability to accurately localize and correctly classify targets. Moreover, mAP also reflects the model’s capacity to reduce false positives and false negatives. Thus, improving mAP directly or indirectly enhances the safety and reliability of object detection systems. For traffic-monitoring applications, AP50 (IoU = 0.5) accommodates scenarios like traffic flow analysis, where minor bounding box deviations are permissible, while AP75 (IoU = 0.75) meets stringent localization requirements for violation capture and autonomous driving perception. The COCO-standard metrics, APs (objects <32×32 pixels), APm (32×32 to 96×96 pixels), and APl (>96×96 pixels), respectively, evaluate small-object recognition in wide-angle surveillance, mid-range event detection, and large-scale vehicle management, providing a complete assessment of model performance across varying object scales in traffic-monitoring scenarios.

In real-world traffic scenarios, especially those involving fixed surveillance cameras or autonomous driving systems, the ability to detect small distant objects, such as pedestrians, cyclists, or motorcycles, is often more critical than precisely classifying their specific type. These objects may be reduced to only a few pixels but represent potential safety risks. The objective is not only to identify “what” the object is, but more importantly, to determine “that” there is an object requiring attention. In this context, even modest improvements in APs can translate to earlier detections, allowing for additional reaction times (e.g., 0.3–0.5 s), which can be crucial in high-speed traffic environments.

Implementation details: These experiments were realized on eight nvidia geforce 2080Ti GPUs and pytorch 1.10, Python 3.8, and cuda 11.7. Since our model and YOLOv8 share most of the backbone network (0–8) and some of the header blocks, there are many weights that can be transferred from YOLOv8 to our proposed model, and by using these weights, we can save a lot of training time.

### 4.3. Object Detection in VisDrone2019

Settings: We train the model on VisDrone for 100 epochs, and the first 3 epochs are used to warm up for training, with the learning rate set to 0.1. The SGD optimizer is used for training, and 0.01 is used as the initial learning rate; the last iteration of the cycle is decayed to 0.01 of the initial learning rate. The SGD momentum is set to 0.8, and the optimizer weight decay is set to 0.0005. Since the input image size is 1536 × 1000, the batch size of the training data volume is set to 8. The results of our model are validated using COCOAPI. For data enhancement, we uses a random left–right flip with a probability of 0.5.

Results: We compare our method with several related methods. The quantitative results are listed in Table 1 and Table 2, the bolded numbers in the table indicate the optimal result among all methods. In the attention module we added, *S* is usually set to 7 as the divisor of the feature map size at each stage. Different sizes of *S* should be selected for input images of different sizes. In this dense prediction task, our *S* is set to 15 to achieve more accurate predictions with the complexity of the regional path and token attention. As the size of the region becomes smaller, the size of *k* is gradually increased to maintain a reasonable number of tokens.

As shown in Table 1, our model outperforms other comparison groups. The performance improvement is significant on small, medium, and large objects. For example, relative to TPH-YOLO, our architecture achieves a performance improvement of 3.14. Compared to CZ Det, our architecture improves the score of small objects more than that of large and medium objects, reaching an astonishing 2.32. This fully illustrates the excellent performance of our new architecture on small-object detection.

We present the visualization results of our method on the VisDrone 2019 test dataset across various scenarios. As shown in Figure 5, Figure 6, Figure 7, Figure 8, Figure 9, Figure 10, Figure 11 and Figure 12, the boxes in the images represent the detected objects, with yellow indicating cars, purple representing trucks, blue for people, cyan for vans, lime for buses, red for bicycles, green for pedestrians, magenta for motorcycles, black for tricycles, and brown for awning tricycles. Red circles highlight the differences between the detection methods. Specifically, Figure 5 and Figure 6 show detection results in high-speed scenarios; Figure 7 and Figure 8 represent scenes with traffic surveillance perspectives; Figure 9 and Figure 10 display nighttime scenes; Figure 11 and Figure 12 illustrate aerial viewpoint scenes. As shown in the red circle in Figure 6, the size of the vehicle in the rear is very small, and our detection method demonstrates excellent performance in this case. In Figure 5a,b,d, rear vehicles are not effectively detected, while in Figure 5c, the blue circle indicates a background misjudgment, leading to a false positive detection. In Figure 8, as shown in the red circle, our method is still able to detect objects with significant occlusion. In the nighttime scenario, as shown in Figure 9a, the ship in the upper part of the image is misclassified as a car, leading to a false positive due to background misjudgment. In Figure 9b,c, a large number of duplicate detections of vehicles in the lower middle part of the image lead to false positives. In Figure 9d, the vehicle on the far left side of the image, moving at high speeds, is not detected. In Figure 12, our method performs better in detecting the pedestrian in the lower-left corner and the small-sized car at the top of the image. The other four methods all exhibit varying degrees of missed detections, leading to false negatives, as shown in Figure 11.

### 4.4. Object Detection in UA-DETRAC

Settings: For the UA-DETRAC dataset, since it provides 100 video files, its frame rate is 25 frames per second. In order to use this dataset, we extract a picture at an interval of 1s for training. Experimental settings set the size of the picture as 960 × 640. Compared with VisDrone, its image size is significantly smaller, and the number of small objects is significantly reduced. We do not report the accuracy with respect to evaluating its large, medium, and small objects. In order to balance training speed and accuracy, *S* is set to 7 to reduce the overall complexity; this is similar to Swin Transformer, which uses a window size of 7. Other settings are consistent with VisDrone.

Results: We compared our method with several related methods on UA-DETRAC. As shown in Table 3, our performance on this dataset exceeds the previous best model. This shows the effectiveness of our hybrid architecture and feature fusion. Specifically, our framework improved by 1.26 percentage points relative to VSTAM and by 3.31, 4.62, and 7.85 percentage points relative to FFAVOD, SpotNet, and CenterNet, respectively.

We present the visualization results of our method and comparison methods on the UA-DETRAC test dataset. As shown in Figure 13, Figure 14, Figure 15, Figure 16, Figure 17 and Figure 18, the boxes in the image represent the detected objects, with green indicating buses, yellow representing cars, and cyan for vans. We first perform detection on the original data and then mark the ignored regions, as indicated by the red hatch boxes in these figures. As shown in Figure 13, for the white van at the bottom of the image, none of (a), (b), (c), or (d) detected it correctly. However, our method achieved better performance, as shown in Figure 14. In Figure 15 and Figure 16, our method demonstrates better detection and higher accuracy in obscured areas compared to the comparison methods, proving that our method has higher robustness. Similarly, in Figure 17, (a–d) show poor detection performance for the van obscured by trees and the small-scale vehicle at the top of the image. Our method, however, performs better in detecting both, as shown in Figure 18.

Although our method achieves promising detection performance, it still suffers from certain false positive and false negative errors. For instance, in Figure 7d, an unidentified object on the road safety island is mistakenly detected as a car, likely because its background shape closely resembles a car roof. Additionally, duplicate detections occur for the black vehicle partially occluded by a streetlight in the lower-right corner, which may result from multi-scale detection layers identifying the same target. In Figure 9d, a false positive appears in the red circle at the lower-right area, where the method misclassifies a car as a truck. Moreover, while pedestrians are detected in Figure 9b,c, they are completely missed in Figure 9d. This discrepancy likely stems from the imbalanced sample distribution in the dataset, where car instances significantly outnumber pedestrian and traffic objects. Notably, while our method detects more targets in Figure 11, it results in classification errors for vehicles in the central road area, specifically misidentifying cars (yellow markers) as vans (cyan markers). This suggests limitations in our classifier’s discriminative capability for similar vehicle categories, and there is still room for further improvement.

### 4.5. Ablation Studies and Complexity Studies

To evaluate the effectiveness of our proposed model and thoroughly analyze the contributions of each component, we conducted ablation studies on the VisDrone dataset, along with comprehensive investigations into computational complexity and parameter counts. As shown in Table 4, the most substantial performance improvement stems from the attention mechanism and its integrated module, which boosts mAP by 4.46% compared to the baseline network, with particularly notable enhancements in smal-object detection. While the feature fusion module also contributes to accuracy improvement, its impact is relatively modest. Furthermore, the introduction of the TFAS module significantly improves small-object detection performance, achieving a 2.59% increase in detection accuracy compared to the model without this module. These results further validate the effectiveness of detecting small objects on larger feature maps. In terms of computational complexity, the model’s total FLOPs amount to 247B, with a parameter count of 168 M (calculated based on input images of 2000×1500 resolution).

### 4.6. Physical Experiment

To validate the effectiveness and feasibility of the proposed method, we conducted a series of physical experiments. These experiments were designed to comprehensively test and evaluate the performance of the algorithm in real-world scenarios, with the results aiding in verifying the proposed algorithm and determining its suitability for practical applications. The experiments utilized the RV1126 embedded development board from Zhengdian Atom and the radar–camera integrated device RC-01 provided by Beijing TransMicrowave Technology Company. The former was used for image acquisition, while the latter was employed to verify the radar’s detection performance. Figure 19a shows a photo of our experimental setup on an overpass during the testing phase, where the equipment was mounted on a tripod and controlled via an external computer. Figure 19b is an enlarged image of the device, and the red box represents the camera module ATK-MCIMX335. This device is capable of outputting images at a resolution of 1080×920. Figure 19c displays the radar–camera-integrated device RC-01, provided by Beijing TransMicrowave Technology Company, Beijing, China.

Figure 20 presents a typical case of joint detection using radar and camera. The calibration of the millimeter-wave radar and camera is performed offline. Firstly, in a controlled laboratory environment, a checkerboard pattern is used to calibrate the intrinsic parameters of the camera, and the intrinsic parameter matrix is obtained using the MATLAB camera calibration toolbox (R2022b). Since the relative position between the camera and the millimeter-wave radar remains fixed in our radar–vision-integrated system, we employ radar angular reflection to enable the camera and radar to capture their respective features and form several sets of one-to-one matching points. The Perspective-n-Point (PnP) algorithm is then used to solve for the extrinsic parameters of the millimeter-wave radar and the camera, thus fulfilling the prerequisite conditions for detection.

The first experiment aimed to validate the effectiveness of our model in real-world scenarios. Video was captured by RV1126, and our model was then applied for object detection. During the testing phase, 500 frames of images were randomly selected from the captured video to evaluate the detection performance of the proposed method. The accuracy was calculated manually, and the results show that using our method, objects can be correctly detected in all images, exceeding 98%. Notably, our model performed exceptionally well with respect to small objects at long distances. Figure 21 shows the detection results of traffic monitoring at an intersection. The targets located in the upper part of the image are far from the camera and therefore appear small in size. The proposed method remains effective in detecting these small distant targets, which is crucial for the early identification of potential hazards or for assessing queue congestion levels.

The second experiment employed our model to evaluate the radar’s detection performance. Camera and radar data were collected using the RC-01 radar–camera-integrated device, which was installed in the same manner as in actual traffic monitoring. Figure 22 presents the test results from a section of the Beijing–Harbin highway. As shown in Figure 22a, the lanes were divided, and the radar only detected vehicles in a single direction, whereas our model detected vehicles in all directions (Figure 22b). From the figure, it is evident that the radar’s detection performance degrades significantly with increasing distance, while the video-based detection remains more stable, especially for small objects that are farther away.

Additionally, we calculated the radar’s detection accuracy, as summarized in Table 5. Within 0~200 m, the detection accuracy was 99.1%; within 200~400 m, it dropped to 87.5%; beyond 400 m, the accuracy further declined to 57.6%. This further demonstrates that with increasing distance, the radar’s detection capability significantly decreases. Although in the well-lit scenario, the radar’s overall object detection performance is not as good as the vision-based camera detection, especially for small and distant targets, the radar still demonstrates irreplaceable advantages in practical applications. Specifically, when the camera fails to detect objects due to issues such as occlusion, strong backlight, rain, fog, or low visibility, radars can still provide object information based on physical signal reflections, effectively alleviating the vulnerabilities of a visual perception system. Therefore, the development of an integrated radar–camera perception system is a key direction for enhancing the reliability and robustness of object detection systems in intelligent transportation scenarios.

## 5. Conclusions

This paper proposes a Multi-Feature Fusion Method based on adaptive dilation convolution to address the challenge of small-object detection in traffic surveillance scenarios. The key contributions of this work include the following: Firstly, a novel fusion network incorporating multi-scale feature fusion modules and triple-feature encoding modules is developed, providing a technical foundation for radar–vision sensor data fusion. Secondly, adaptive dilation convolution dynamically adjusts dilation rates to reduce intra-class size variance in feature maps, thereby enhancing feature representation for small objects. Thirdly, the SBRA (Skipped Binary-Routing Attention) module is introduced into the backbone network to improve contextual modeling through global dependency learning.

The proposed multi-feature fusion method based on adaptive dilated convolution demonstrates not only clear technical advantages through experimental validation but also significant practical value in real-world applications such as smart city systems. Specifically, the method achieves mAP improvements of 1% on the UA-DETRAC dataset and 3% on the VisDrone dataset, reflecting its practical effectiveness in complex scenarios. In smart city traffic management, the method significantly enhances the ability of surveillance cameras to detect small objects in urban environments, thereby supporting traffic signal scheduling, improving traffic flow, and enabling the early warning of potential accidents. Furthermore, the effectiveness of the approach has been validated through physical experiments, highlighting its real-world applicability and its potential contribution to the advancement of intelligent transportation systems.

While the proposed method demonstrates promising performances, several limitations remain: The model shows room for improvement in distinguishing similar categories (e.g., cars vs. vans) in complex backgrounds or densely populated scenes. Although multi-sensor fusion partially mitigates the impact of lighting and weather conditions, detection performances may degrade under extreme conditions such as heavy rain or dense fog.

Future research directions include exploring further fusion strategies for radar and camera data to enhance detection robustness in complex environments. Additionally, optimizing computational efficiency through techniques like knowledge distillation and neural network pruning will make the method more suitable for deployment on edge computing devices. Reducing reliance on fine-grained annotated data and leveraging unlabeled or weakly labeled data will help improve the model’s generalization ability.

## Figures and Tables

**Figure 1 sensors-25-03182-f001:**
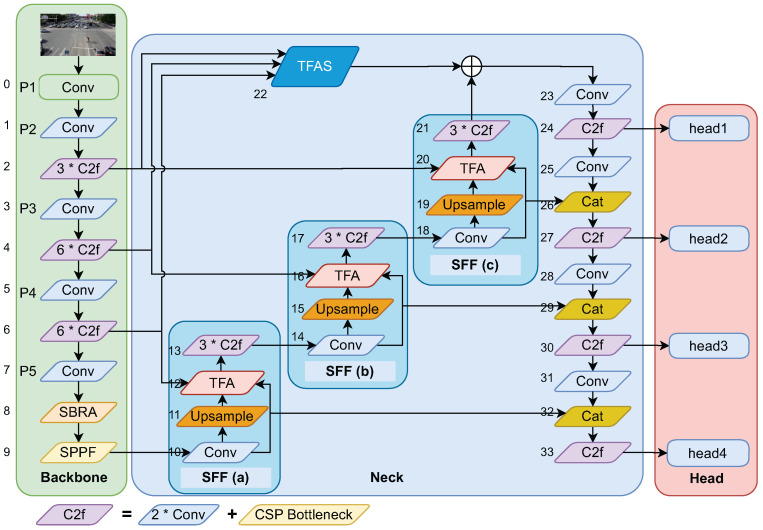
The overall architecture. Images are input to a (a) CSPDarkNet53 backbone network to extract features, and the connection between features is established through the terminal SBRA module. Then, through the SPPF module, (b) Neck uses a structure similar to PANet to aggregate features from different backbone network levels. Among them, TFA is a fusion structure built using adaptive dilated convolution. (c) Four prediction heads use the feature map after TFA in Neck to predict boxes at four different scales.

**Figure 2 sensors-25-03182-f002:**
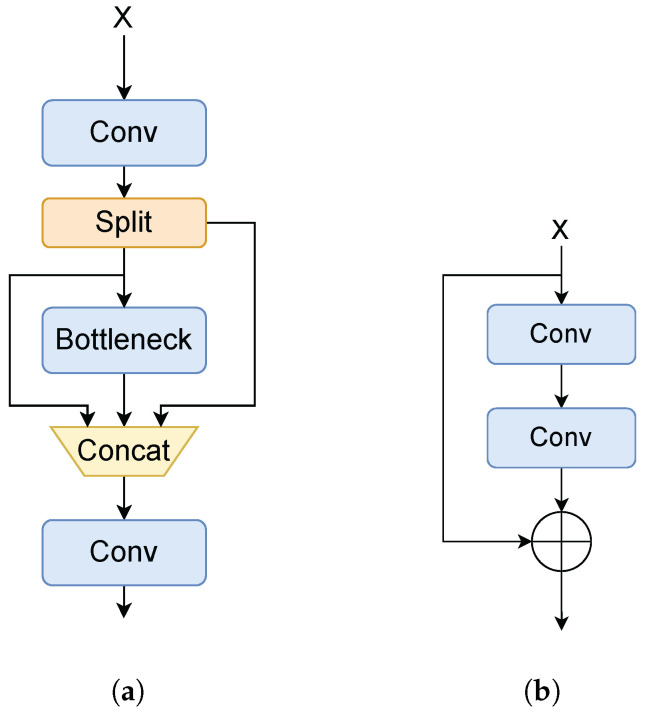
The architecture of C2f and Bottleneck. (**a**) is the C2f module, and (**b**) is the Bottleneck module. It should be noted that the Bottleneck in Backbone and Neck’s C2f is different. The latter has no residual branch.

**Figure 3 sensors-25-03182-f003:**
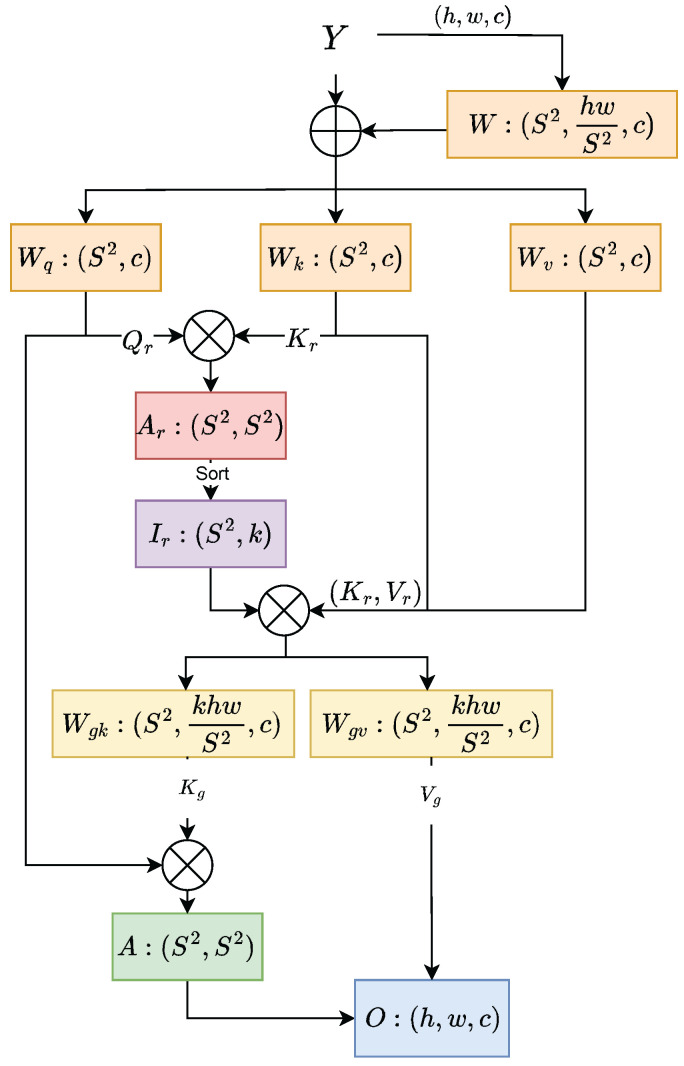
The architecture of SBRA. *S* represents the square root of the number of regions; *h* and *w* represent the height and width of the input feature map; *c* represents the number of channels; *k* is a constant representing the number of regions with the highest correlation. $A_{r}$ represents the adjacency matrix of the region, $I_{r}$ represents the index in the adjacency matrix, and $W_{gk}$ and $W_{gv}$ represent the key–value pairs after the index. A represents region-to-region attention, and ⨁ and ⨂ represent element-wise summation and matrix multiplication, respectively.

**Figure 4 sensors-25-03182-f004:**
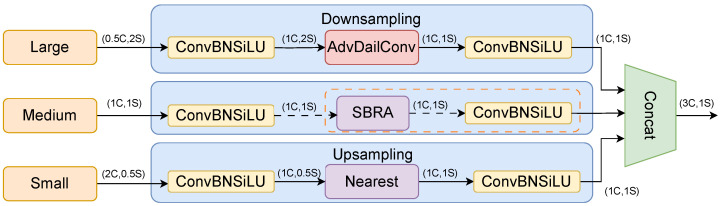
Feature fusion module of adaptive dilated convolution and attention, where C represents the number of channels and S represents the size of the feature map. ConvBNSiLU is the Conv, batch normalization, and SiLU operation, and Nearest represents nearest neighbor interpolation. The input of each module is three feature maps of different sizes. The part surrounded by the dotted line is only used in the TFAS module in Figure 1 but not in the TFA module.

**Figure 5 sensors-25-03182-f005:**
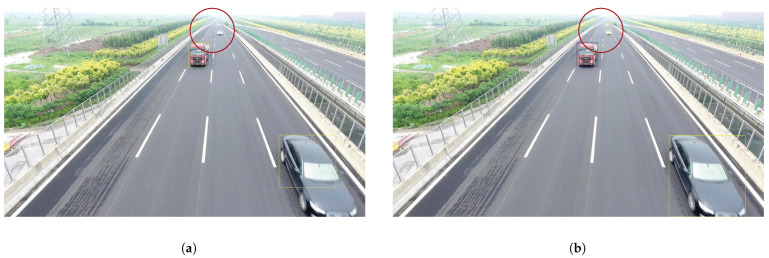
Visualization results of different object detection methods in highway scenarios on the VisDrone test set. Different categories are displayed in distinct colored boxes, with the key differences between the methods highlighted by red and blue circles. (**a**) is the detection result of YOLOv8x. (**b**) is the detection result of DEIM. (**c**) is the detection result of the CZ Det method. (**d**) is the detection result of TPH-YOLO.

**Figure 6 sensors-25-03182-f006:**
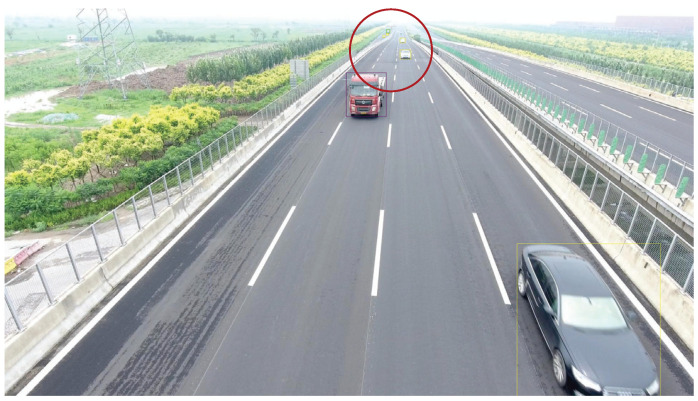
Visualization result of our methods in highway scenarios on the VisDrone test set. Different categories are represented by boxes of different colors, and the main differences with other methods are marked by the red circles.

**Figure 7 sensors-25-03182-f007:**
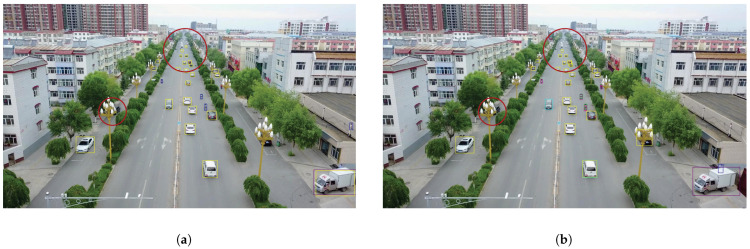
Visualization results of different object detection methods in scenes similar to traffic monitoring on the VisDrone test set. Different categories are displayed in distinct colored boxes, with the key differences between the methods highlighted by red circles. (**a**) is the detection result of YOLOv8x. (**b**) is the detection result of DEIM. (**c**) is the detection result of the CZ Det method. (**d**) is the detection result of TPH-YOLO.

**Figure 8 sensors-25-03182-f008:**
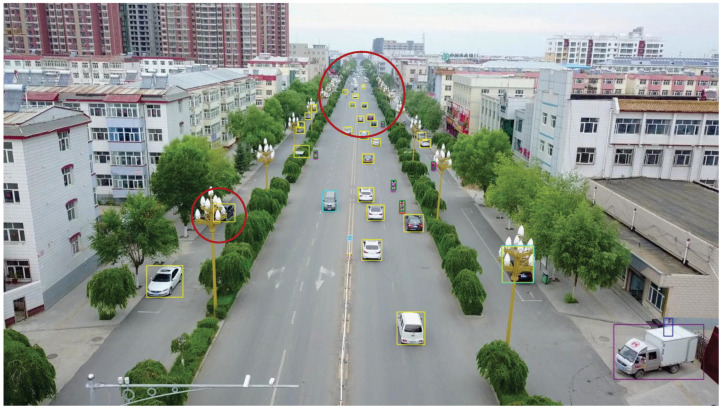
Visualization result of our method in similar traffic-monitoring scenarios on the VisDrone test set. Different categories are represented by boxes of different colors, and the main differences with other methods are marked by the red circles.

**Figure 9 sensors-25-03182-f009:**
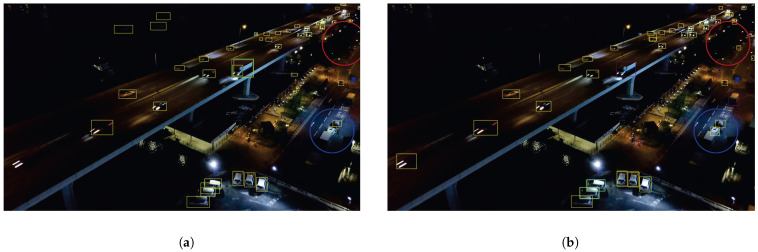
Visualization results of different object detection methods in night scenes on the VisDrone test set. Different categories are displayed in distinct colored boxes, with the key differences between the methods highlighted by red and blue circles. (**a**) is the detection result of YOLOv8x. (**b**) is the detection result of DEIM. (**c**) is the detection result of the CZ Det method. (**d**) is the detection result of TPH-YOLO.

**Figure 10 sensors-25-03182-f010:**
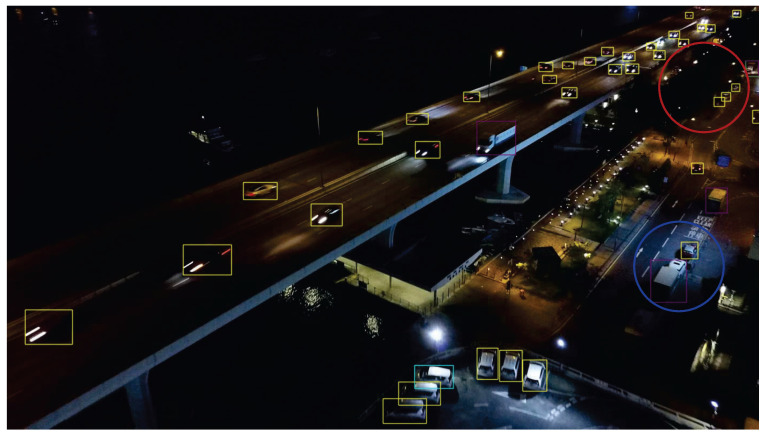
Visualization result of our method in night scenes on the VisDrone test set. Different categories are represented by boxes of different colors, and the main differences with other methods are marked by the red and blue circles.

**Figure 11 sensors-25-03182-f011:**
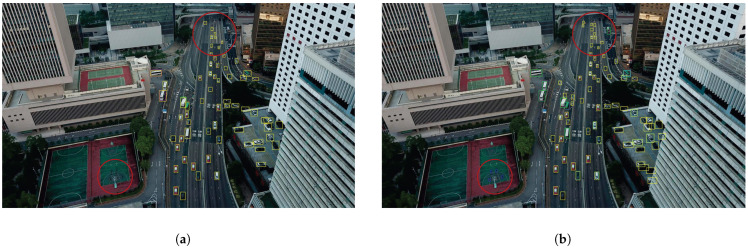
Visualization results of different object detection methods from an overhead perspective on the VisDrone test set. Different categories are displayed in distinct colored boxes, with the key differences between the methods highlighted by red circles. (**a**) is the detection result of YOLOv8x. (**b**) is the detection result of DEIM. (**c**) is the detection result of the CZ Det method. (**d**) is the detection result of TPH-YOLO.

**Figure 12 sensors-25-03182-f012:**
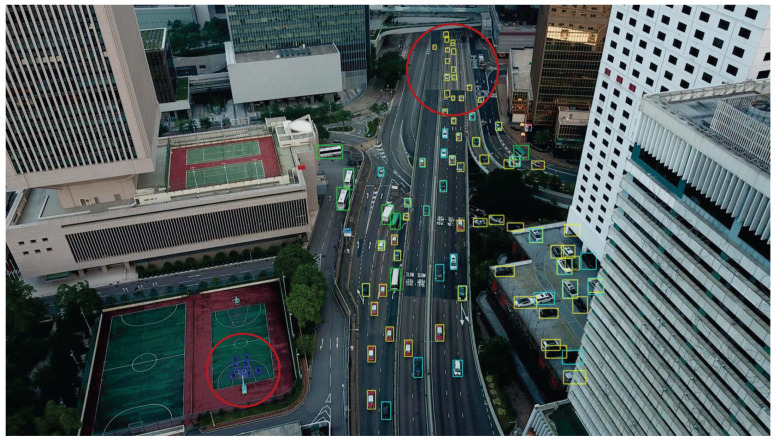
Visualization result of our method from an overhead perspective on the VisDrone test set. Different categories are represented by boxes of different colors, and the main differences with other methods are marked by the red circles.

**Figure 13 sensors-25-03182-f013:**
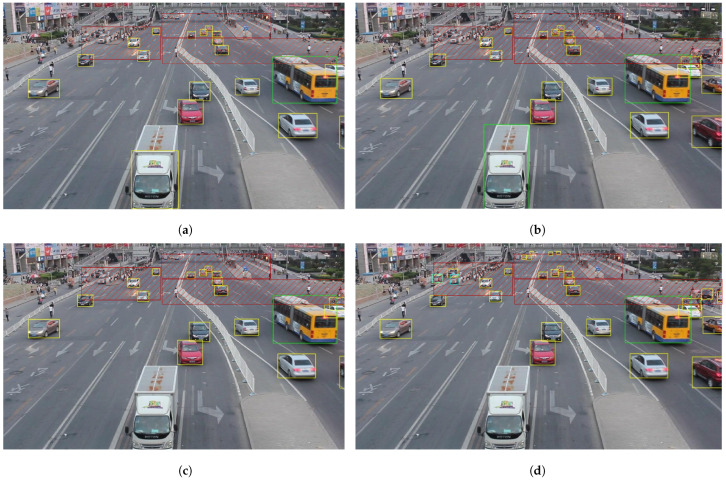
Visualization results of different object detection methods in the MVI40711 scenario on the UA-DETRAC test set. Different categories are represented by boxes of different colors, and the red slash boxes indicate ignored areas in the dataset. (**a**) is the detection result of CenterNet. (**b**) is the detection result of SpotNet method. (**c**) is the detection result of FFAVOD. (**d**) is the detection result of VSTAM.

**Figure 14 sensors-25-03182-f014:**
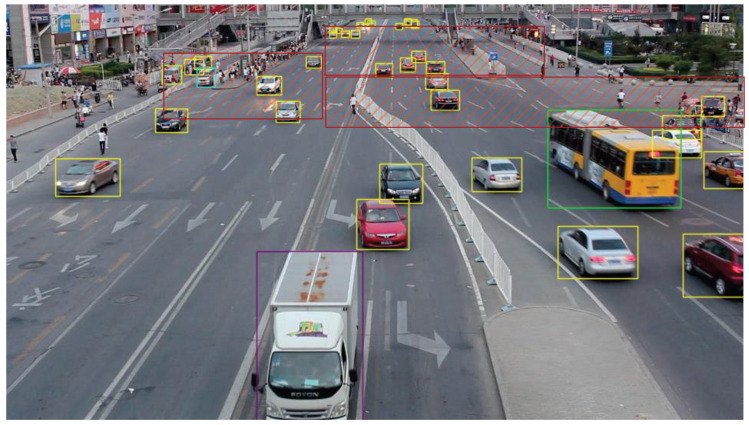
The detection results of our method on the UA-DETRAC test set in the MVI40711 scenario. Different categories are represented by boxes of different colors, and the red slash boxes indicate ignored areas in the dataset.

**Figure 15 sensors-25-03182-f015:**
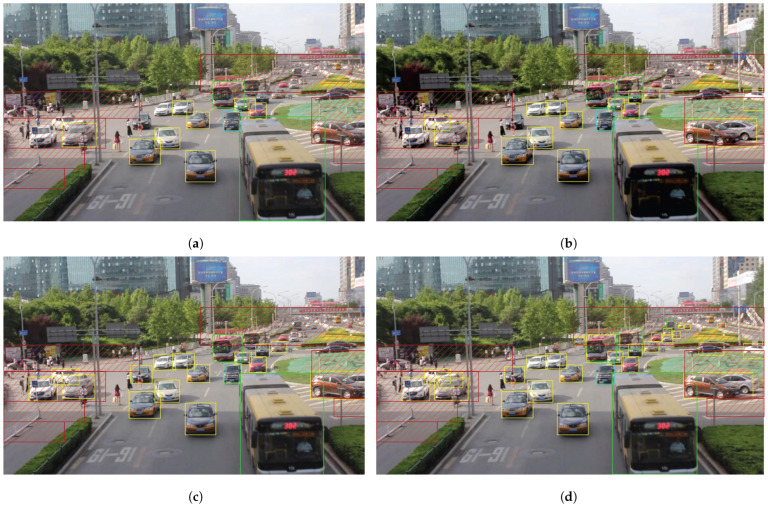
Visualization results of different object detection methods in the MVI39311 scenario on the UA-DETRAC test set. Different categories are represented by boxes of different colors, and the red slash boxes indicate ignored areas in the dataset. (**a**) is the detection result of CenterNet. (**b**) is the detection result of SpotNet method. (**c**) is the detection result of FFAVOD. (**d**) is the detection result of VSTAM.

**Figure 16 sensors-25-03182-f016:**
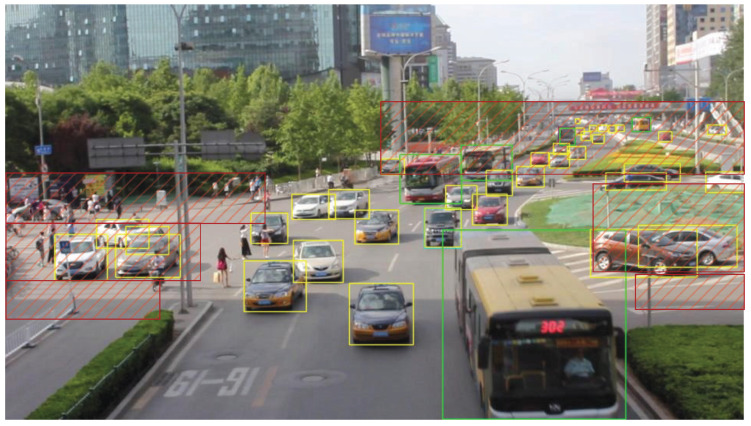
The detection results of our method on the UA-DETRAC test set in the MVI39311 scenario. Different categories are represented by boxes of different colors, and the red slash boxes indicate ignored areas in the dataset.

**Figure 17 sensors-25-03182-f017:**
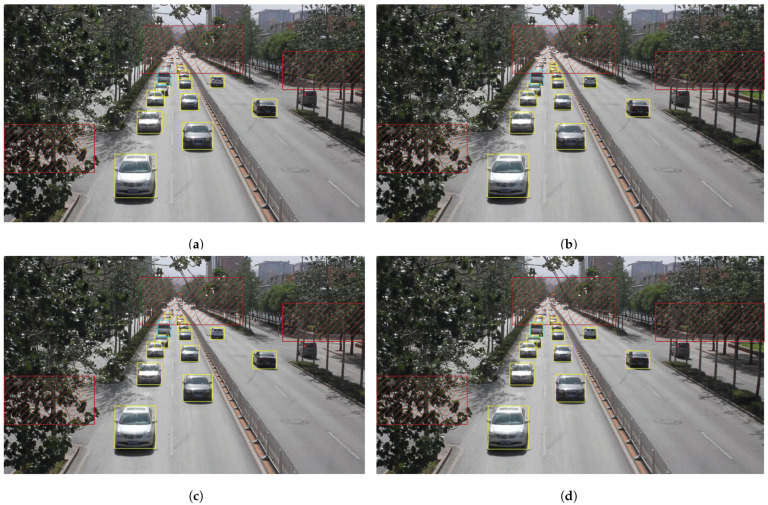
Visualization results of different object detection methods in the MVI39031 scenario on the UA-DETRAC test set. Different categories are represented by boxes of different colors, and the red slash boxes indicate ignored areas in the dataset. (**a**) is the detection result of CenterNet. (**b**) is the detection result of SpotNet method. (**c**) is the detection result of FFAVOD. (**d**) is the detection result of VSTAM.

**Figure 18 sensors-25-03182-f018:**
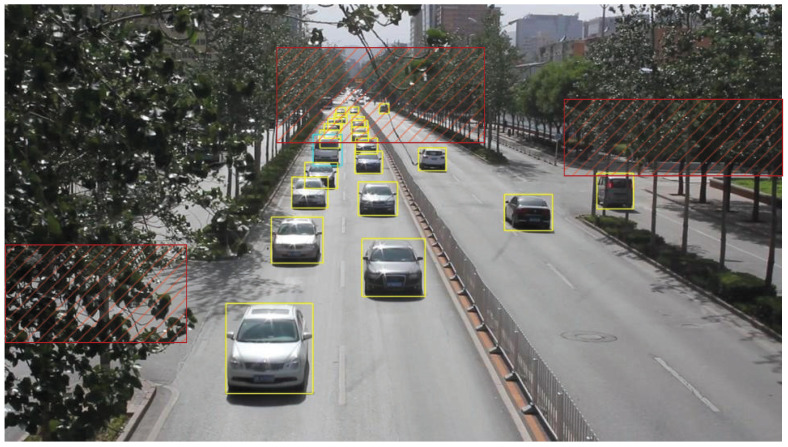
The detection results of our method on the UA-DETRAC test set in the MVI39031 scenario. Different categories are represented by boxes of different colors, and the red slash boxes indicate ignored areas in the dataset.

**Figure 19 sensors-25-03182-f019:**
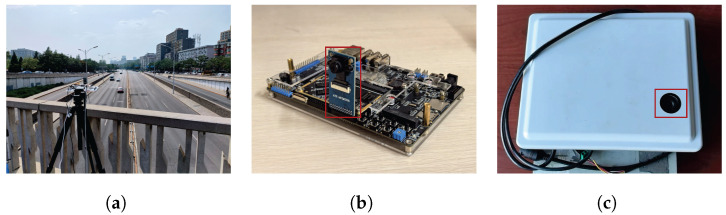
Experimental scenes and equipment. (**a**) is a photo of the experimental scene, (**b**) is a detailed picture of the experimental equipment, and (**c**) is a radar–camera-integrated device. The red boxes in (**b**,**c**) indicate the camera.

**Figure 20 sensors-25-03182-f020:**
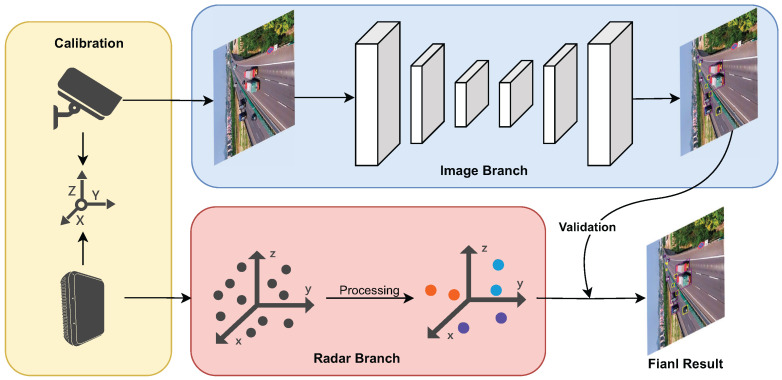
Schematic diagram of the radar and camera system.

**Figure 21 sensors-25-03182-f021:**
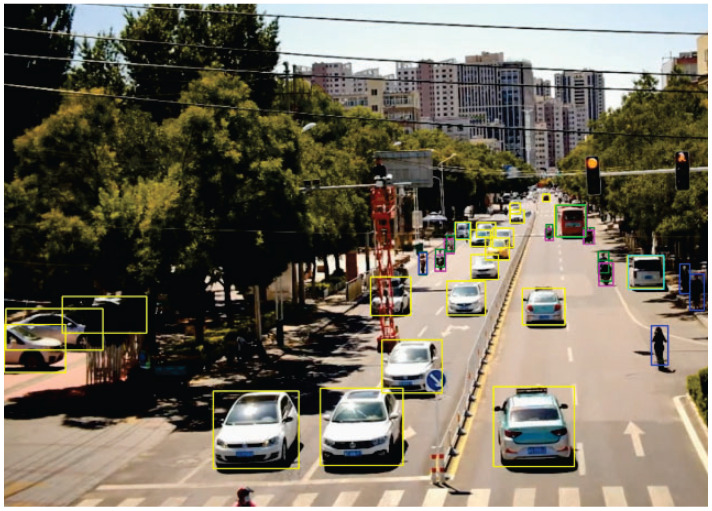
The detection results of traffic surveillance at an intersection. Different categories are represented by boxes of different colors.

**Figure 22 sensors-25-03182-f022:**
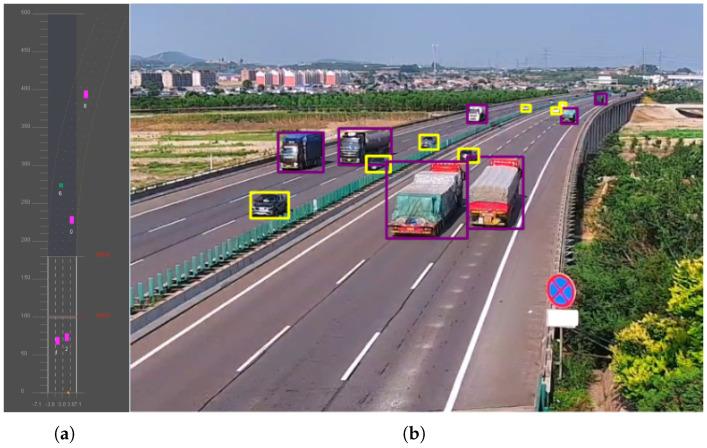
The detection results from a screenshot of the recorded video on the Beijing to Harbin highway. Different categories are represented by boxes of different colors.

**Table 1 sensors-25-03182-t001:** Comparison of object detection tasks based on VisDrone2019 test. The bolded numbers indicate optimal results.

Methods	mAP	AP_50_	AP_75_	AP_*s*_	AP_*m*_	AP_*l*_
YOLOv8-x6	29.12	46.91	30.78	27.16	32.19	35.48
DEIM	29.50	49.54	30.99	20.31	41.95	**55.10**
CZ Det.	33.13	57.45	32.78	26.05	42.17	43.24
TPH-YOLO	35.74	57.31	-	-	-	-
Ours	**38.88**	**59.12**	**37.25**	**28.37**	**43.38**	44.52

**Table 2 sensors-25-03182-t002:** Comparison of object detection tasks in all classes based on VisDrone2019 test. The bolded numbers in the table indicate the optimal results.

Methods	All	Pedestrian	People	Bicycle	Car	Van	Truck	Tricycle	Awning Tricycle	Bus	Motor
YOLOv8-x6	29.12	21.57	13.32	11.83	52.75	46.33	38.43	21.91	18.74	50.56	23.45
DEIM	32.45	22.47	12.03	11.97	57.34	46.52	41.33	22.42	18.94	56.43	25.47
CZ Det.	33.13	24.25	12.57	12.04	61.72	46.58	42.24	23.31	19.56	57.69	26.31
TPH-YOLO	37.21	28.87	16.76	15.54	68.93	**50.19**	45.09	27.32	23.67	**63.27**	30.58
Ours	**38.88**	**29.93**	**18.44**	**16.32**	**69.81**	49.76	**46.32**	**27.87**	**24.21**	61.78	**31.45**

**Table 3 sensors-25-03182-t003:** Comparison of object detection tasks based on the UA-DETRAC test dataset. The bolded numbers in the table indicate the optimal results.

Methods	mAP	AP_0.5_	AP_0.75_
CenterNet	83.52	96.46	91.23
SpotNet	86.78	96.72	91.38
FFAVOD	88.06	97.87	91.75
VSTAM	90.26	98.13	92.41
Ours	**91.37**	**98.45**	**92.57**

**Table 4 sensors-25-03182-t004:** Ablation studies and complexity studies performed based on the VisDrone2019 test. The bolded numbers in the table indicate the optimal results.

Methods	mAP	AP_50_	AP_75_	AP_*s*_	AP_*m*_	AP_*l*_	Params	FLOPs
baseline	29.12	46.91	30.78	21.16	32.19	35.48	78M	165B
+SBRA	33.58	54.29	34.11	24.36	36.15	39.27	113M	198B
+SBRA+TFA	35.72	58.31	36.47	25.78	38.64	42.58	136M	225B
+SBRA+TFA+TFAS	**38.88**	**59.12**	**37.25**	**28.37**	**43.38**	**44.52**	168M	247B

**Table 5 sensors-25-03182-t005:** The accuracy of radar detection obtained based on image detection results.

Distance (m)	AP	Car	Truck	Bus
0<d⩽200	99.1	99.2	98.7	99.4
200<d⩽400	81.5	81.3	80.8	82.3
d>400	57.6	57.7	55.1	59.4

## Data Availability

The original datasets are publicly available from the VisDrone dataset (https://github.com/VisDrone/VisDrone-Dataset) (accessed on 14 May 2025) and the UA-DETRAC Benchmark Suite (https://www.albany.edu/cnse/research/computer-vision-machine-learning-lab) (accessed on 14 May 2025).

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
