# Peer review of "Multi-Feature Fusion Method Based on Adaptive Dilation Convolution for Small-Object Detection"

_sensors, 2025, doi:10.3390/s25103182_

Round 1
Reviewer 1 Report
Comments and Suggestions for Authors
The paper presents a novel small object detection method that integrates self-attention mechanisms and a multi-scale feature fusion network, specifically applied to traffic monitoring scenarios. The proposed approach enhances the accuracy and robustness of small object detection on YOLOv8 by incorporating self-attention mechanisms. Additionally, the multi-scale feature fusion network dynamically adjusts the convolutional kernel size, improving the detail representation of small objects and capturing more contextual information. Experimental results demonstrate that the method achieves superior performance improvements on both the UA-DETRAC and VisDrone datasets compared to existing methods. However, the overall writing quality of the paper is subpar, and the authors should consider a comprehensive revision of the manuscript. Below are detailed comments and suggestions for improvement:
- The proposed method is based on camera-based techniques. Therefore, the discussion on radar in the introduction should be reduced or omitted. The abstract should also be revised accordingly. The related work section in the introduction is overly lengthy and lacks a critical discussion of the limitations of existing methods. It is recommended to retain only a few representative works, briefly introduce them, and highlight their limitations (i.e., the problems addressed by this paper). The remaining content should be moved to Section 2 (Related Work). The challenges faced in small object detection are not clearly explained. Specifically, the statement in lines 134-135, "This requires the detector to have enhanced receptive field adaptation capabilities and dynamic receptive fields," needs to be supported by relevant citations. This background is crucial to the core inspiration of the paper and should be elaborated in detail.
2.The authors mention, "During the processing, the adaptive dilation convolution is introduced, where the feature map is divided according to different frequency bands, and then different dilation rates are used for different frequency bands to dynamically adjust the receptive field." This should not be presented as an independent contribution. Instead, the authors should emphasize the novelty of applying adaptive dilation convolution in the context of small object detection.
3.The authors should reduce the discussion on radar in the early sections of the paper. Additionally, the extensibility of the proposed method should be thoroughly discussed in the experiment section.
4.The use of related terminology should be precise. For example, "adaptive dilated convolution" should be used accurately and consistently throughout the paper.
5.In the visualization section, the differences in the images should be highlighted with zoomed-in markers to make the comparisons clearer.
6.The conclusion section is currently disorganized and reads more like an introduction. It is recommended to restructure this section and include a discussion on the limitations of the proposed method.
7.While the paper presents a promising approach to small object detection, significant revisions are required to improve the clarity, organization, and overall quality of the manuscript. The authors should address the above comments and consider a major rewrite to ensure the paper meets the standards of the journal.
Comments on the Quality of English Language1.The use of related terminology should be precise. For example, "adaptive dilated convolution" should be used accurately and consistently throughout the paper.
Reviewer 2 Report
Comments and Suggestions for Authors
The authors present a new algorithm for detecting small objects. The algorithm itself is of some interest, but to my deepest regret I cannot recommend the work for publication in the Sensors journal. Because in my opinion there is not even a reasonable statement of the problem, as a result we have "research for the sake of research" without any practical significance.
Of course, there are many tasks that require identification of objects (including small objects). And each of them has its own requirements for the accuracy of identification and even its own metrics. And, by the way, its own datasets for verification.
It is worth saying that method A is superior to method B if method B does not provide acceptable metric values, and method A does. For example, for a safe takeoff of an airplane, it is necessary to detect 99.9999% of approaching birds in time. Method A detects 80% of birds, method B 70%. Does this mean that method A is better than method B? No, they are both unacceptable. At the same time, for some other (ornithological) task, method A may actually be better. But we must first define the task, and not just compare abstract numbers. It is good, of course, that the algorithm is slightly better at distinguishing a bus from a pedestrian, but what benefit does this provide to people? Can it be used in some kind of road safety systems? The article does not say a word about this.
In its current form, the work is quite suitable for some purely mathematical journal not about sensorics. Of course, even in this case, a number of points need to be improved:
1) Serious proofreading is required. The work is full of typos ("fianl", "dilateion", "servel" etc) and even errors (such as "more lower" in line 60 and "the lanes was divided" in line 625)
2) Many abbreviations are not deciphered. At the same time, for some reason, a number of never-reused abbreviations have been introduced (such as FOD in line 236)
3) The pictures are not presentable at all. There is not even information about which colors correspond to which classes of objects. In picture 6c, the necessary immediately catches the eye; in the others, one has to look very closely for a long time to find important differences.
4) The figures in lines 10-11 are given with a clear excess of accuracy. 1% and 3% would be quite sufficient.
Comments on the Quality of English LanguageIn principle, everything that the authors wanted to say is clear. But many constructions with three or more subordinate clauses and the constant use of the passive voice create a certain "cumbersomeness".
Reviewer 3 Report
Comments and Suggestions for Authors
This study proposes a method for small object detection in traffic scenarios by integrating attention mechanisms and adaptive dilation convolutions within a hybrid YOLOv8-based architecture. The approach is of interest for scientific community. However, several aspects require refinement before the manuscript can be considered for publication. The following comments could help the authors improve their work:
- In the introduction chapter, it is necesary clarify more explicitly the novelty of the proposed approach compared to previous methods, emphasizing how it overcomes the typical challenges of small object detection in different aspects of transport engineering. Furthermore, expand the description of the transformations applied to the feature maps at each stage of the network to provide a clearer understanding of the proposed architecture.
- It is necesary in the Methodology chapter explain how key hyperparameters, such as dilation rates for different frequency bands) are selected and justify their relevance to the final performance. In addition, it is necesary include more detailed information about the datasets used, particularly regarding the diversity of scenarios and conditions (e.g., lighting, traffic density) they cover.
- Related to the application, it is necesary discuss the limitations and potential biases of the methodology, such as possible failure cases involving extreme occlusion or uncommon object types. Furthermore, it is recomemded provide a dedicated section addressing computational complexity and memory consumption to assess the scalability and feasibility of the proposed method.
- It is recomended to compare the proposed method against a broader range of baseline algorithms, including recent unified detectors and attention-based approaches, to better contextualize performance. In addition, justify the choice of evaluation metrics such as AP, AP50, APs and explain how they relate to real-world performance in traffic monitoring scenarios.
- It is recommended to include an error analysis section discussing the most common types of false positives and false negatives, supported by illustrative examples. In addition, I suggest confirm whether the experiments cover a sufficient variety of conditions such as weather, time of day, and camera angles, to strengthen the external validity of the findings.
- Suggest future research directions, such as integrating additional sensors or adapting the method to non-traffic environments.
- Strengthen the discussion of practical implications by connecting the empirical results to real-world applications, such as smart cities or autonomous systems.
- Consider including a technical appendix detailing the radar-camera calibration or validation procedures, highlighting key parameters and results.
Comments on the Quality of English LanguageN/A
Round 2
Reviewer 1 Report
Comments and Suggestions for Authors
This paper presents a novel network integrating self-attention and multi-scale feature fusion, specifically designed for traffic surveillance scenarios. The proposed method enhances the accuracy and robustness of small object detection in YOLOv8 through the incorporation of self-attention.
Key Contributions:
- The work introduces a new feature fusion architecture based on YOLOv8 that effectively integrates multi-level features from sensors.
- The paper presents an adaptive dilated convolution approach to reduce intra-class size variations in feature maps while achieving dynamic receptive field adaptation.
- The network employs self-attention to capture global dependencies, enabling superior contextual analysis.
Strengths:
- Experimental results demonstrate the method's effectiveness in improving small object detection performance.
- The approach shows excellent practical performance in real-world validation.
- The authors have made substantial improvements to the manuscript quality: (1) Significant restructuring for improved readability; (2) Clear articulation of network design motivations; (3) Comprehensive experimental comparisons with state-of-the-art methods
Recommendation: I appreciate the substantial improvements made to the manuscript since its previous submission. The proposed method, integrating self-attention with multi-scale feature fusion for small object detection, is both practical and beneficial for real-world traffic surveillance applications. I recommend acceptance of the current revised version.
Reviewer 2 Report
Comments and Suggestions for Authors
The revised version is undoubtedly better than the original one.
However, in my opinion, it is still unclear in which sensor applications the results of the work can be useful.
Let's take my example with birds again. If we are considering the problem of preventing a plane from colliding with a bird, we generally do not care whether it is a crane or a heron. But if we are ornithologists, then we need to distinguish them. That is, even the division of objects into equivalence classes is entirely determined by the original problem. Then metrics and criteria are determined. And they also depend on the original problem.
And one can't just take the "abstract task of identifying objects in the test dataset", typical metrics AP50 and mAP and compare them. Well, one can, but there is no practical benefit from this, in my opinion. And you got a gain in metrics in the third significant digit, what will this give me as a road user (driver or pedestrian)? Will traffic lights work better? Will there be fewer road accidents? Or will nothing change at all?
With all due respect to the authors, the work still requires significant revision. First of all, in terms of the formulation of the problem (everything else is just a consequence).
Reviewer 3 Report
Comments and Suggestions for Authors
All my comments have been satisfactorily addressed. Congratulations to the authors.
